# Clinical and Epidemiological Profile of Patients with Invasive Aspergillosis from a Fourth Level Hospital in Bogota, Colombia: A Retrospective Study

**DOI:** 10.3390/jof7121092

**Published:** 2021-12-18

**Authors:** Ana Goyeneche-García, Juan Rodríguez-Oyuela, Guillermo Sánchez, Carolina Firacative

**Affiliations:** 1Group MICROS Research Incubator, School of Medicine and Health Sciences, Universidad del Rosario, Bogota 111221, Colombia; ana.goyeneche@urosario.edu.co (A.G.-G.); juanes.rodriguez@urosario.edu.co (J.R.-O.); 2Fundación Cardioinfantil, Bogota 110131, Colombia; efrain.sanchez@urosario.edu.co; 3Studies in Translational Microbiology and Emerging Diseases (MICROS) Research Group, School of Medicine and Health Sciences, Universidad del Rosario, Bogota 111221, Colombia

**Keywords:** *Aspergillus*, aspergillosis, epidemiology, Colombia

## Abstract

Invasive aspergillosis (IA) is a severe mycosis caused by *Aspergillus* species. The infection mainly affects immunocompromised patients with a significant clinical burden. This study aimed to determine the clinical and epidemiological characteristics of patients diagnosed with IA in a fourth level hospital in Colombia, as these data are scarce in the country. A retrospective, observational study, from a single center was conducted with 34 male and 32 female patients, between 1 month- and 90-year-old, diagnosed with proven (18.2%), probable (74.2%) and possible (7.6%) IA, during a 21-year period. The most frequent underlying conditions for IA were chemotherapy (39.4%) and corticosteroid use (34.8%). The lung was the most common affected organ (92.4%). Computed tomography (CT) imaging findings were mainly nodules (57.6%) and consolidation (31.8%). A low positive correlation was found between serum galactomannan and hospitalization length. *Aspergillus fumigatus* prevailed (73.3%) in sputum and bronchoalveolar lavage cultures. Most patients were hospitalized in general wards (63.6%) and treated with voriconazole (80.3%). Mortality rate was 15.2%. Common risk factors for IA were identified in the Colombian cohort, including medications and underlying diseases. However, their frequency differs from other countries, reinforcing the idea that local surveillance is essential and at-risk patients should be carefully monitored.

## 1. Introduction

Invasive aspergillosis (IA) is the most severe and one of the main clinical forms of aspergillosis, an opportunistic mycosis caused by species of the genus *Aspergillus*, a group of saprotrophic filamentous fungi that primarily inhabits the soil. *Aspergillus fumigatus*, which is the most common etiological agent, causes around 90% of clinical cases in the world [1]. In addition, *Aspergillus flavus*, *Aspergillus terreus*, *Aspergillus niger*, and *Aspergillus nidulans*, among others, have also been identified as human pathogens [2]. Although infection with any of these species starts by inhaling spores that circulate in the environment and are transmitted through the air, *Aspergillus* spores that reach the lungs of immunocompetent humans or animals are generally eliminated by neutrophils and macrophages of the innate immune system [3]. However, depending on the virulence or pathogenicity of the species or strain, as well as on the host’s immune status and lung structure and function, *Aspergillus* can lead to a variety of allergic reactions and infectious diseases in immunocompromised individuals, which can progress from the respiratory system to a disseminated or invasive lethal infection [1].

Globally, more than 300,000 cases of IA are estimated to occur per year, with mortality rates exceeding sometimes 90% of cases [4]. Regarding the host, it has been estimated that the incidence of IA is 5 to 25% in patients with acute leukemia, 5 to 10% in patients with allogeneic bone marrow transplantation and 0.5 to 5% in patients with cytotoxic treatment of hematological diseases, autologous bone marrow transplantation and solid organ transplantation (SOT). Among organ recipients, the incidence is highest in heart and lung transplant patients (19 to 26%) followed by liver, pancreas, and kidney transplants (1 to 10%) [1,5]. Additionally, IA is considered an emerging mycosis in patients in intensive care units (ICU) and postoperative patients, mainly due to the dispersion of spores through hospital ventilation systems, as well as in other non-traditional at-risk groups, including patients with chronic lung diseases, with AIDS and those receiving immunomodulatory drugs [6,7,8].

In Colombia, it has been estimated that almost 3000 cases of IA occur per year, from which 13% of cases are related to organ transplant recipients, especially those undergoing stem cell transplants [9]. Case reports with infrequent *Aspergillus* species or uncommon clinical presentations of aspergillosis have been also reported in Colombia [10,11,12,13,14], as well as aspergillosis in children [14,15,16,17], and reports of the presence of *Aspergillus* spores around hospitals, which increases the risk of patients developing aspergillosis [18,19]. However, there are no epidemiological studies with complete data on the occurrence of this mycosis in the country, which can serve to validate the estimated incidence. Additionally, in Colombia, surveillance of most mycoses, including aspergillosis, is not mandatory, and therefore many epidemiological data are missing or underestimated.

Apart from the common risk factors to develop IA, which have been very well studied in developed countries, clinical presentations, patients characteristics, and epidemiology of the disease seem to vary locally and new at-risk populations are being identified, especially in developing countries [7]. In this way, decisions about which strategy to select for correctly managing *Aspergillus* infections will depend not only on the access to rapid and accurate diagnostics, but also on patient characteristics and local epidemiology. In this study, the demographics, clinical manifestations, imaging, laboratory and mycological findings, treatment, and mortality of patients with IA from a fourth level hospital in Bogota, Colombia were retrospectively evaluated. The analysis of these variables is useful to identify the patients that may have a higher risk of developing IA in the studied center, allowing improved management and outcomes. As reports on the epidemiology of aspergillosis in Colombia are scarce, this study provides real data that contribute, regionally, nationally, and internationally, to the study of IA, one of the most lethal clinical forms of aspergillosis.

## 2. Materials and Methods

### 2.1. Study Population

Data from patients diagnosed with IA between January 2000 and December 2020 at the Fundacion Cardioinfantil, a major fourth level hospital in Bogota, Colombia, was gathered. Cases were identified by searching for the following International Classification of Diseases (ICD)-10 codes, which are registered in the medical records of patients: B44 (Aspergillosis), B44.0 (Invasive pulmonary aspergillosis), B44.1 (Other pulmonary aspergillosis), B44.7 (Disseminated aspergillosis), B44.8 (other forms of aspergillosis), B44.9 (Aspergillosis, unspecified), and B49.X (Unspecified mycosis). IA was classified as proven, probable, and possible according to the European Organization for Research and Treatment of Cancer and the Mycoses Study Group Education and Research Consortium (EORTC/MSGERC) criteria [20]. Demographic characteristics from patients included gender, age, height, weight, and body mass index (BMI) (Table 1). Host factors, clinical manifestations, imaging findings on chest computer tomography (CT) scan, treatment and outcome were recorded for all patients. Microbiological evidence, including serum and bronchoalveolar lavage (BAL) galactomannan (GM) test (Platelia™ *Aspergillus* Ag, Bio-Rad Laboratories, Hercules, CA, USA), as well as sputum or BAL culture were performed for most patients and test results were recorded. A serum or BAL GM optical density index (ODI) ≥ 0.5 was considered positive for IA [21,22]. When GM test was performed more than once in the same patient, the highest OD was considered for the analysis. Laboratory test results at admission were registered for most patients.

### 2.2. Statistical Analyses

Data are shown as numbers and percentages. For quantitative variables, median and range are shown. Spearman’s rank correlation coefficient was used to assess linear dependence between quantitative variables. Correlation was judged very strong from 1 to 0.9, strong from 0.9 to 0.7, moderate from 0.7 to 0.5, low from 0.5 to 0.3 and poor from 0.3 to 0. A Mann–Whitney nonparametric test was performed for unpaired samples. The alpha risk was set to 5% (α = 0.05). Statistical analysis was performed with the online platform EasyMedStat (version 3.9; www.easymedstat.com, accessed on 12 November 2021). A graph with the distribution of cases of IA identified per year was generated using the software GraphPad Prism v 7.05. 

### 2.3. Ethics

Patient identifiable data were not used or disclosed in this study and were kept anonymous during the whole data analysis. The ethics committee from Fundacion Cardioinfantil approved the study under the protocol CEIC-4304-2020.

## 3. Results

In a 21-year period, a total of 66 patients diagnosed with IA were identified at Fundacion Cardioinfantil. From these, 55 patients (83.3%) were diagnosed in the last decade (2011 to 2020) (Figure 1). In the first three studied years and in 2007, no cases were identified. Of all the cases, 12 (18.2%), 49 (74.2%) and five (7.6%) were classified respectively as proven, probable, and possible IA. The demographic and clinical characteristics of all patients are summarized in Table 1. In the studied population, 34 (51.5%) patients were men and 32 (38.5%) were women, between 1 month- and 90-year-old, with an average age of 43.6 years. Two patients were neonates, 12 were between the ages of 7 and 16, and 52 were adults (18 years and older). Of adult patients, 16 (30.2%) were overweight and two (3.8%) were underweight. Chemotherapy was the most common underlying condition among the patients (26 out of 66, 39.4%). Corticosteroid use was another major predisposing factor for IA (23 out of 66, 34.8%), with almost half of the patients receiving prednisolone (47.8%), followed by dexamethasone (17.4%). Neutropenia at the time of diagnosis was found in 11 patients (16.7%), all receiving chemotherapy and seven of those with leukemia. Hypertension, diabetes mellitus and acute leukemia were found as main underlying diseases in IA patients (24.2%, 19.7% and 18.2%, respectively). Heart transplant recipients prevailed among transplanted patients with IA. Other less common underlying conditions were also reported (Table 1). Twenty-seven patients (40.9%) presented three or more known risk factors for IA [6].

Detailed clinical characteristics of all patients are shown in Table 2. The lung was the most frequent affected organ (61 out of 66, 92.4%), with nodules being the main finding by CT scan (57.6%), followed by consolidation (31.8%). Multiple lesions (63.6%) were more common than single lesions (36.4%), with 17 patients (25.8%) having at least three lesions on thoracic CT imaging.

In four cases of pulmonary aspergillosis, cerebral aspergillosis was suspected, as in these patients, nodules in the brain were observed. Another patient with pulmonary aspergillosis presented as well with infection in the paranasal sinuses, which showed obstruction and inflammation in a CT scan, and typical *Aspergillus* hyphae in the biopsy. One patient developed secondary cutaneous lesions, from hematogenous spread of fungi, with typical hyphae of *Aspergillus* observed in skin biopsies. In another patient, *Aspergillus* sp. was isolated from the peritoneal fluid. The mediastinum, specifically the trachea, was involved in two cases.

Patients were mostly hospitalized in general wards (63.6%), stayed in hospital an average of 38 days, including patients who had died, and were treated with voriconazole alone (47%) or in combination with one or more antifungals (33.3%). Even though 22 out of the 66 patients presented septic shock (33.3%), mortality rate was low (15.2%). 

A positive serum GM was observed in 31 out of 49 patients (63.3%), and a positive BAL GM was observed in 19 out of 23 patients (82.6%). When cultures were performed, growth of *Aspergillus* was less frequent from sputum (25.4%) than from BAL (31.3%), with *A. fumigatus sensu lato* being the prevalent species, recovered in 22 out of 30 positive cultures (73.3%). Histopathology was conducted in 24 patients with a positive rate of 45.8%. Routine laboratory results varied among patients (Table 3).

A low positive correlation was found between serum GM ODI and the length of stay in hospital (ρ = 0.33; r^2^ = 0.009; *p* = 0.021). No other variable correlated with the duration of hospitalization (*p* > 0.05). Patients with positive and negative serum GM did not differ in the neutrophils and leukocytes count nor in the creatinine or blood urea nitrogen levels (BUN) (*p* > 0.05). GM levels did not correlate with the number of CT scan findings. Comparisons between groups of patients with proven, probable, and possible IA were not carried out, due to the small number of cases.

## 4. Discussion

Globally, the amount of IA cases continues to increase every year with the growing number of at-risk patients. However, in Colombia, epidemiological data about this opportunistic mycosis is at the moment scarce and the burden of the disease in the country remains underestimated. This study is, to our knowledge, the first retrospective investigation that provides clinical data and characteristics on the epidemiology of patients diagnosed with IA in a major hospital in Bogota, the capital city of Colombia.

In the group of studied patients, chemotherapy was the most common risk factor for IA, accounting for almost 40% of patients, together with patients receiving corticosteroids, which accounted for about 35% of cases. Exogenous and severe immunosuppression, often related to chemotherapy and corticosteroid use, results in prolonged neutropenia, which is still the single main risk factor for invasive infections caused by *Aspergillus* [1,8]. Diabetes mellitus and acute leukemia, together accounting for almost 40% of cases, which are known to increase the risk of patients to develop IA, were the most frequently reported underlying diseases, although the frequencies slightly differ from those reported in other countries [7,23].

Considering that Fundacion Cardioinfantil is a recognized transplant center, performing an average of 100 transplant surgeries per year [24], it was possible to determine that in SOT patients, the highest incidence for IA was found in lung recipients (9.4%), followed by heart (5%), liver (0.4%) and kidney (0.3%) transplanted patients. Although IA cases in SOT recipients were only identified in the last decade of the study, the incidence values are comparable with estimated incidences reported in other Latin American countries such as Brazil [25], Chile [26], Ecuador [27], Peru [28], and Mexico [29], as well as in larger transplant centers in the USA [30]. Even though, hematopoietic stem cell transplantation (HSCT) recipients have been recognized for several decades as the group of patients with the higher risk to develop aspergillosis [6], patients undergoing HSCT were the less frequently group developing IA in our center (4.5%), as these patients are typically referred from other institutions.

As reported elsewhere, the incidence of IA and other forms of aspergillosis is increasing in nonneutropenic patients and in patients with other underlying diseases apart from diabetes and leukemia [6,31,32]. In this study, chronic obstructive pulmonary disease (COPD), chronic kidney disease and tuberculosis, were each found in more than 10% of patients with IA. Although the occurrence of IA in patients with COPD has been associated with prolonged use of corticosteroids [33,34], five of the eight patients with COPD in our center were not treated with these drugs, yet four of them presented concomitant tuberculosis and one concomitant asthma. As occurring in other developing countries with a high incidence of tuberculosis, IA and mycobacterial infection was infrequently found in our institution (10.6%) [7]. Other underlying conditions, less commonly reported in *Aspergillus* infections, such as autoimmune diseases, coronary artery disease, pulmonary hypertension, asthma, cirrhosis, and cystic fibrosis, were found in our study in patients with IA. However, these conditions were always accompanied by at least one additional predisposing factor, predominantly corticosteroids use. Contrasting, in patients with IA and HIV, the viral infection was the only underlying disease. Even though this fungal-viral concomitance is uncommon due to little or no compromise of neutrophil function in HIV infection, IA in HIV patients is associated with poor survival, probably because of the advanced AIDS and immunosuppression [6,7,35]. Hence, in HIV positive population, prompt recognition of the fungal infection and appropriate treatment are critical.

Regarding the clinical characteristics of our cohort, it was found that pulmonary aspergillosis was by far the most common form of infection, independently of the underlying condition of the patients. Thus, the lung was involved in 93.9% of cases, of which one case had a concomitant infection of the sinuses and four cases a concomitant infection of the brain, as cerebral lesions were observed in those patients, thus neurological infection was inferred [21]. Aspergillosis is mainly acquired after inhalation of conidia, as such most patients present sinopulmonary tract involvement, and while hematogenous dissemination to extrapulmonary sites may occur, these manifestations are infrequent [36]. In patients with IA from this study, mediastinum, and peritoneum involvement as well as secondary skin lesions in a neutropenic woman were also found, which even though rare, have been previously reported in both immunocompromised and immunocompetent people [37,38,39,40]. While classic radiological findings associated with IA such as nodules, consolidation, ground-glass opacity, the halo sign, and cavitary lesions, were often observed in our patients, pleural effusions, tree-in-bud pattern, and other less common CT scan findings, were also found. Imaging findings remain however nonspecific and indistinguishable from other lung infections, which can lead to misdiagnosis, especially in nonneutropenic or low-risk patients [31,41]. It is known, as well, that among ICU patients, the typical CT scan pattern suggestive of invasive fungal disease is not observed in about 70% of the cases, leading to an incorrect diagnosis and a delay in optimal therapy [42].

Even though mortality rate in IA patients can exceed 90% of cases [4], most of the patients in our study survived (84.8%). In the 21-year studied period, the total number of identified cases, which in none of the years surpassed 9 cases, together with the mortality rate, were strongly influenced by the number of immunocompromised patients or with predisposing factors that are admitted to our center. In addition, in our institution, IA may not be recognized in patients with non-classical risk factors or with nonspecific clinical symptoms, which leads to an underestimated occurrence of the disease [31,43]. Not only the overall epidemiological data suggest that the incidence of IA in non-haemato-oncological and non-HSCT patients is miscalculated, but also IA is among the most frequently missed diagnoses, according to post-mortem examinations [1,6,33,44].

Despite the evidence for using serum and BAL GM in making an early diagnosis of IA, detection of GM antigen was not always performed in our cohort. However, when measurements were available, positivity rate for this marker was 63.3% in serum and 82.6% in BAL. As reported previously, serum GM sensitivity appears to be lower in patients other than those with hematologic malignancy and HSCT, as those that predominate in our hospital, decreasing to about 20% in SOT recipients [21,22]. In the other hand, while BAL GM was more sensitive than BAL culture, the number of cultures positive for *Aspergillus* was higher from BAL (31.3%) than from sputum samples (25.4%), with *A. fumigatus* being the most common isolated species from all samples. Although BAL GM has a sensitivity that exceeds 70%, which is in turn higher compared with BAL culture, the specificity is rather low, as the lungs are often colonized by *Aspergillus* [21]. Together with GM detection, lung biopsy examination has an important impact in the diagnosis of IA [1,21]. Considering however that biopsies are often difficult and risky to undertake, in our cohort, only 24 patients (36.4%) underwent this procedure, with a positive rate that did not exceed 50% of cases.

Constant surveillance of IA is therefore of paramount importance, as this mycosis is considered one of the most expensive diseases in the hospital setting, due to the length of hospitalization, the complications of patients and the costly treatment [45]. In our cohort, patients stayed in hospital an average of 38 days, either in general wards or in ICU, and voriconazole, which has one of the highest prices among antifungal agents [46], was the most commonly used treatment, either as monotherapy or in combination with other antimycotics. Regular surveillance of cases of invasive infections caused by molds, such as *Aspergillus*, is also strongly recommended in transplant centers, such as our institution, as if an increase in the incidence or the occurrence of these infections is observed, the hospital environment source should be prompted identified [21].

This study not only contributed with epidemiological data on the incidence of IA in Colombia and globally, but also aids recognizing the characteristics of the patients receiving care in our institution and affected by this invasive fungal infection. Knowing the clinical and epidemiological profiles of IA patients is essential to be able to consider early interventions and preventive measures in certain groups of patients that should be carefully monitored due to their propensity to develop this mycosis. Early diagnosis and immediate institution of antifungal therapy may as well help reducing complications and thus mortality. Finally, invasive mycoses such as IA remain major threats to patients receiving SOT, hindering the success of the transplantation, which is of particular importance as our institution is one of the main transplant centers in Colombia and Latin America.

## Figures and Tables

**Figure 1 jof-07-01092-f001:**
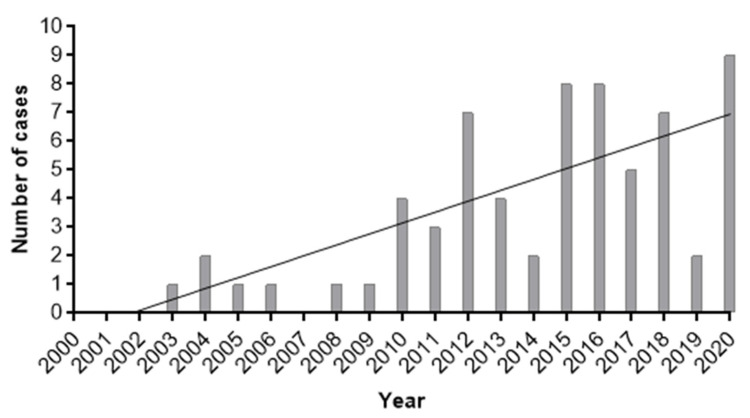
Distribution of cases of invasive aspergillosis per year, which seems to increase during time.

**Table 1 jof-07-01092-t001:** Demographic characteristics and underlying conditions of patients diagnosed with invasive aspergillosis.

Characteristic of Patients (*n* = 66)	Number (%)
Gender	
Male	34 (51.5)
Female	32 (48.5)
Underlying condition	
Chemotherapy	26 (39.4)
Corticosteroids use	23 (34.8)
Hypertension	16 (24.2)
Diabetes mellitus	13 (19.7)
Leukemia	12 (18.2)
Acute myeloid	6 (9.1)
Acute lymphoblastic	6 (9.1)
Neutropenia	11 (16.7)
Solid organ transplantation	10 (15.2)
Heart	4 (6.1)
Lung	3 (4.5)
Liver	2 (3.0)
Kidney	1 (1.5)
Chronic obstructive pulmonary disease	8 (12.1)
Chronic kidney disease	7 (10.6)
Tuberculosis	7 (10.6)
Autoimmune disease	6 (9.1)
Coronary artery disease	6 (9.1)
Pulmonary hypertension	5 (7.6)
Asthma	4 (6.1)
Cirrhosis	4 (6.1)
Cystic fibrosis	4 (6.1)
Human immunodeficiency virus	4 (6.1)
Hematopoietic stem cell transplant	3 (4.5)
	**Median (Range)**
Age (years)	47 (1 month–90)
Body mass index (kg/m^2^)	22.1 (10.3–33.5)

**Table 2 jof-07-01092-t002:** Clinical characteristics of patients diagnosed with invasive aspergillosis.

Characteristic of Patients (*n* = 66)	Number (%)
Site of infection	
Lung ^a^	60 (90.9)
Paranasal sinuses ^b^	2 (3.0)
Mediastinum	2 (3.0)
Peritoneum	1 (1.5)
Skin	1 (1.5)
Computed tomography scan findings	
Nodules	38 (57.6)
Consolidation	21 (31.8)
Ground-glass opacity	18 (27.3)
Halo sign	13 (19.7)
Cavity	11 (16.7)
Pleural effusion	7 (10.6)
Tree-in-bud pattern	6 (9.1)
Mycetoma	4 (6.1)
Other	7 (10.6)
Hospitalization	
General ward	42 (63.6)
Intensive Care Unit	24 (36.4)
Treatment ^c^	
Voriconazole	53 (80.3)
Liposomal amphotericin B	13 (19.7)
Itraconazole	9 (13.6)
Other	19 (28.8)
Complications	
Septic shock	22 (33.3)
Outcome	
Alive	56 (84.8)
Dead	10 (15.2)
	**Median (Range)**
Hospital length of stay (days)	32.5 (3–150)
Time to death after admission (days)	40 (12–114)

^a^ Four cases with possible concomitant brain infection. ^b^ One case concomitant with lung infection. ^c^ Usage of the antifungal alone or in combination.

**Table 3 jof-07-01092-t003:** Mycological and laboratory findings of patients diagnosed with invasive aspergillosis.

Characteristic	Number (%)
Mycological findings	
Serum GM ≥ 0.5 (*n* = 49)	31 (63.3)
BAL GM ≥ 0.5 (*n* = 23)	19 (82.6)
Sputum culture (*n* = 59)	15 (25.4)
BAL culture (*n* = 48)	15 (31.3)
*Aspergillus* species (*n* = 30)	
*Aspergillus fumigatus*	22 (73.3)
*Aspergillus flavus*	5 (16.7)
*Aspergillus niger*	2 (6.7)
*Aspergillus* sp.	1 (3.3)
Biopsy (*n* = 24)	11 (45.8)
**Test**	**Median (Range)**
Neutrophils (*n* = 66)	4085 (0–34,180)
Leukocytes (*n* = 66)	6225 (40–56,800)
Creatinine (mg/dL) (*n* = 65)	0.8 (0.2–11.2)
BUN (mg/dL) (*n* = 63)	14 (3–64)
INR (*n* = 57)	1.08 (0.90–1.83)
Prothrombin time (sec) (*n* = 57)	13.1 (10.1–23.6)
AST (U/L) (*n* = 54)	28 (10–264)
ALT (U/L) (*n* = 54)	38 (7–214)
Bilirrubin (mg/dL) (*n* = 53)	0.5 (0.2–26.2)
Procalcitonin (ng/mL) (*n* = 18)	0.3 (0.05–7.47)

GM, galactomannan; BAL, bronchoalveolar lavage; BUN: blood urea nitrogen; INR, international normalized ratio; AST, aspartate aminotransferase; ALT, alanine aminotransferase. n, number of cases where tests were performed.

## Data Availability

Data from this study are available from the corresponding author upon request.

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
