# Peer review of "Clinical and Epidemiological Profile of Patients with Invasive Aspergillosis from a Fourth Level Hospital in Bogota, Colombia: A Retrospective Study"

_jof, 2021, doi:10.3390/jof7121092_

Round 1
Reviewer 1 Report
The manuscript describes the clinical features and underlying risk factors of patients diagnosed with invasive aspergillosis (IA) in a quaternary level hospital in Colombia. Overall the manuscript is well presented, and although there are previous publications describing cohorts of IA patients in other South American countries, as the authors state there appears to be a paucity of published data from Colombia itself. Only 4.5% of the patients in this cohort were HSCT patients, which may explain the relatively low mortality rate. The study is limited by its retrospective nature, relatively small sample size and the fact it is a single centre analysis.
I have some specific comments as follows:
Suggest authors clarify how cases of IA were identified in their institution. Was this through a coding system or was a prospectively updated database of these cases kept? How likely was it that cases may have been missed?
Line 80: presumably "AI" should be "IA"
Line 94: GM ODI >0.5 was considered positive for IA, but presumably for the designation of proven/probable/possible disease the EORTC criteria GM ODI cut offs were used which are higher than 0.5. Suggest clarify this in the manuscript.
Line 108: suggest amending to eg "patient identifiable data were not used," as clearly patient data were used during the analysis.
I don't think the regression line or r squared value add much to figure 1 and would suggest removing these from the figure.
Line 142/table 2: please clarify whether or not all extra pulmonary disease was confirmed on histological examination of biopsy or if these diagnoses were made on the basis of radiological abnormalities. Also please clarify which exact anatomical site in the "mediastinal" cases were affected.
Please describe range of time post admission that the IA related deaths occurred.
Line 158 -9: suggest authors clarify if length of stay data used in this analysis included patients who had died. Ideally the individual data points should be displayed in a figure.
Did some patients have more than serum (or BAL) GM test during their admission? If so, have the authors reported the first GM test for each patient or the GM test with the highest optical index?
The first paragraph of the discussion appears to have been included in error.
Author Response
Thank you for the very helpful comments to our manuscript. Please find below our detailed answers to the reviewer's comments.
The manuscript describes the clinical features and underlying risk factors of patients diagnosed with invasive aspergillosis (IA) in a quaternary level hospital in Colombia. Overall the manuscript is well presented, and although there are previous publications describing cohorts of IA patients in other South American countries, as the authors state there appears to be a paucity of published data from Colombia itself. Only 4.5% of the patients in this cohort were HSCT patients, which may explain the relatively low mortality rate. The study is limited by its retrospective nature, relatively small sample size and the fact it is a single centre analysis.
Answer: The authors thank the reviewer for his/her comments.
I have some specific comments as follows:
Suggest authors clarify how cases of IA were identified in their institution. Was this through a coding system or was a prospectively updated database of these cases kept? How likely was it that cases may have been missed?
Answer: In lines 86-90, it was specified how the IA cases were identified in the hospital. It was through a coding system (International Classification of Diseases (ICD)-10). As we also included, unspecified mycosis, it is very unlikely that we missed some cases.
Line 80: presumably "AI" should be "IA"
Answer: Thanks for noticing this. The change was done accordingly.
Line 94: GM ODI >0.5 was considered positive for IA, but presumably for the designation of proven/probable/possible disease the EORTC criteria GM ODI cut offs were used which are higher than 0.5. Suggest clarify this in the manuscript.
Answer: In our institution, an OD>0.5 is considered positive, following the instructions of the Platellia Aspergillus Ag (Bio-Rad Laboratories). In addition, as referenced in the manuscript (Ref. 21, 22), the “Guidelines for the Diagnosis and Management of Aspergillosis: 2016 Update by the IDSA” states “As the optimal threshold for GM positivity has not been determined; an OD of 1.0 has been cleared by the FDA for clinical testing, although some experts consider positivity at OD > 0.5. A higher threshold OD index results in a lower sensitivity but a higher specificity.” Considering this, we used OD > 0.5 as positive for IA.
Line 108: suggest amending to eg "patient identifiable data were not used," as clearly patient data were used during the analysis.
Answer: The change was done accordingly.
I don't think the regression line or r squared value add much to figure 1 and would suggest removing these from the figure.
Answer: The r squared value was removed from the figure legend. However, the regression line was kept, as it shows the trend that the number of cases increase per year.
Line 142/table 2: please clarify whether or not all extra pulmonary disease was confirmed on histological examination of biopsy or if these diagnoses were made on the basis of radiological abnormalities. Also please clarify which exact anatomical site in the "mediastinal" cases were affected.
Answer: In line 154-159, it was specified how the extra pulmonary cases were confirmed and the exact affected anatomical site in the mediastinum.
Please describe range of time post admission that the IA related deaths occurred.
Answer: The time to death after admission was included in Table 2.
Line 158 -9: suggest authors clarify if length of stay data used in this analysis included patients who had died. Ideally the individual data points should be displayed in a figure.
Answer: In line 161, it was mentioned that length of stay in hospital included patients who dad died. Due to the small number of cases, a figure was not included.
Did some patients have more than serum (or BAL) GM test during their admission? If so, have the authors reported the first GM test for each patient or the GM test with the highest optical index?
Answer: We reported the GM test with the highest optical index. This was clarified in the text in lines 101-102.
The first paragraph of the discussion appears to have been included in error.
Answer: Thank you very much for noticing this. The whole paragraph was deleted.
Reviewer 2 Report
Dear Authors:
thank you very much for this article.
Nice work, easy to follow.
Did you identified as species level Aspergillus fumigatus isolates? They were Aspergillus section funigati? or were A. fumigatus s.s??
In those patients in whom the GM test was performed; did they always use the same brand / kit?
Did you use the LFD-aspergillus or any other POC in any of those patients?
kind regards
Author Response
Thank you for the very helpful comments to our manuscript. Please find below our detailed answers to the reviewer's comments.
Dear Authors:
thank you very much for this article. Nice work, easy to follow.
Answer: The authors thank the reviewer for his/her comments.
Did you identified as species level Aspergillus fumigatus isolates? They were Aspergillus section funigati? or were A. fumigatus s.s??
Answer: They were Aspergillus fumigatus sensu latu. This is now stated in the manuscript in line 172.
In those patients in whom the GM test was performed; did they always use the same brand / kit?
Answer: Yes, the Platellia Aspergillus Ag from BioRad was always used. This is now stated in the manuscript in line 98.
Did you use the LFD-aspergillus or any other POC in any of those patients?
Answer: No. Only the Platelia test is available in our institution